# Pixel-Coordinate-Induced Human Pose High-Precision Estimation Method

**Xuefei Sun** †, **Mohammed Jajere Adamu** †, **Ruifeng Zhang, Xin Guan** * **and Qiang Li** *

School of Microelectronicss, Tianjin University, Tianjin 300072, China
* Correspondence: guanxin@tju.edu.cn (X.G.); liqiang@tju.edu.cn (Q.L.)
† These authors contributed equally to this work.

**Abstract:** Accurately estimating human pose is crucial for providing feedback during exercises or musical performances, but the complex and flexible nature of human joints makes it challenging. Additionally, traditional methods often neglect pixel coordinates, which are naturally present in high-resolution images of the human body. To address this issue, we propose a novel human pose estimation method that directly incorporates pixel coordinates. Our method adds a coordinate channel to the convolution process and embeds pixel coordinates into the feature map, while also using coordinate attention to capture position- and structure-sensitive features. We further reduce the network parameters and computational cost by using small-scale convolution kernels and a smooth activation function in residual blocks. We evaluate our model on the MPII Human Pose and COCO Keypoint Detection datasets and demonstrate improved accuracy, highlighting the importance of directly incorporating coordinate location information in position-sensitive tasks.

**Keywords:** human pose estimation; high-precision keypoint detection; coordinate convolution; coordinate attention; position sensitivity

## 1. Introduction

Two-dimensional human pose estimation is an important and challenging problem in computer vision, which has wide application in human motion recognition, human–computer interaction, virtual reality, video surveillance, and human trajectory tracking [1,2]. Human pose estimation depends on detecting human body keypoints (such as head, wrists, elbows, knees, and ankles) in a single image or video. The detected keypoints' skeleton structure contains the most information about the human pose. However, due to the complexity and high flexibility of human joints, as well as differences in the integrity of human body parts in different-quality images, videos, and other factors, human keypoint detection remains challenging in various application scenes, such as vision-based motion pose, musical instrument playing pose capture and correction, virtual human motion generation, and human motion data acquisition in exoskeleton robots. High precision is still required to meet the demands of these applications.

The conventional pose estimation methods are based on component models for inference, which generally include pictorial structure and deformable part models [3,4]. By designing human component detectors that use fixed pictorial structure models to identify objects in images and establish the connectivity of each part, we can apply human-body-related constraints for continuous optimization. This approach can help us to better detect and understand the components of the human body in images. The features extracted by the traditional method mainly rely on the Histogram of Oriented Gradients and scale-invariant feature transform, which cannot fully use the image spatial location information and are subject to different appearances, perspectives, and occlusion in the image. Furthermore, the scope of application is limited due to the single structure of the component model, which can only accurately identify basic poses such as upright, sitting, and striding and is less effective for complex human actions.

Due to the excellent feature representation performance, human pose estimation is now more often adopted by deep learning methods. Convolutional neural networks can be used to detect the keypoints of the human body. This approach not only provides richer semantic feature information (such as keypoint categories) but also enables the extraction of multi-scale and multi-type human joint point feature descriptions and interrelationships. Using neural networks, more accurate and robust features can be extracted than artificial ones. A more accurate topology between each joint point in a human pose can be established, facilitating the prediction of more complex poses. The convolutional neural network approach has been dramatically improved in its estimation accuracy and is gradually becoming the mainstream approach to solving human pose estimation.

Two-dimensional multi-person pose estimation methods can be roughly classified into two categories, bottom-up [5–13] and top-down [14–21], where the former first detects all human skeletal joint points and then combines the joint points into multiple human poses. The latter first detects all human body bounding boxes in the image and then estimates the human pose within each box individually. Compared with the bottom-up approaches, the top-down approach first determines the location of the human body. It uses the coordinate position information to locate the joint points in the region where the human body is located, which is beneficial to further improve the estimation accuracy.

In the current top-down approaches, most models thoroughly explore various fusions of the semantic, global, and local detail information of images in extracting features used to characterize the keypoints of the human body, characterizing the structure of the components of the human body through feature information, establishing relationships among the parts from a macroscopic perspective, and extracting local detail information at different scales. For example, the Convolutional Pose Machines (CPM) [15] uses serialized convolutional neural networks to learn texture information while acquiring structural relationships at a distance by expanding the receptive field. Moreover, each stage repeats the generation of confidence maps containing spatial contextual information to guide feature learning in subsequent stages to learn the relative positional relationships of the keypoints of the human body. The Multi-Stage Pose Network (MSPN) [19] enables the repetitive overlapping of high-resolution and low-resolution features, fuses semantic and detailed feature information, and aggregates features from adjacent stages to enhance the characterization capability of feature information and highlight target features. The High-Resolution Network (HRNet) [20] network fuses semantic and channel information at different scales and can always maintain the high-resolution image size to retain the original location information of the whole image. The multi-branch information exchange allows the output high-resolution features to contain more information but not directly utilize the human image pixel location information. Although various existing feature fusion methods further enhance the feature learning ability related to human keypoints, none directly utilize the more important pixel location information implied in the image. Pixel location can provide a relative coordinate system, and the pixel location is relatively unchanged when the image size changes, which is beneficial to maintain the location information of the features and thus improve the accuracy of keypoint estimation.

In addition, existing networks are complex and offer many different forms of convolutions and connections for purposes such as highlighting key features, but reusing extracted features multiple times may cause a certain degree of information redundancy and waste computational resources. For example, the Stacked Hourglass [16] uses multiple identical hourglass modules, each containing symmetric upsampling, downsampling, and cross-layer linking. However, the model effect does not improve significantly with the increase in modules. The Cascaded Pyramid Network (CPN) [18] contains a multi-level network with a complex cascading pyramid structure, and then it requires secondary networks to calibrate further and correct the initial prediction results. We need to conduct more in-depth research to determine whether the use of pyramid connections for feature extraction is a wasteful use of computational resources, and whether the correction effect of secondary networks can be balanced with resource consumption. HRNet [20] contains

many branches to fuse multi-scale information, and there is also some doubt about whether feature duplication will exist in multi-branch feature extraction. Due to the complex links and too many branches, it is difficult to determine the contribution of a specific part of the structure to the feature representation with similar accuracy, so there is still much room for research to reduce the complexity of the model, reduce the unnecessary connections of the network, and improve the computational efficiency.

This paper proposes a novel pixel coordinate guided estimation model to solve the above problems. The model reduces the network parameter amount while maintaining high estimation accuracy. The model cleverly utilizes the characteristics inherent in the image and incorporates the high-precision pixel coordinate position information hidden in the high-resolution human pose image into the network structure design, thereby significantly improving the model performance and verifying the effectiveness of directly combining coordinate position information. A top-down method is used to detect the human part of the image first, and then the coordinate position information is directly combined with learning of the keypoint features of the human body. The network design and training process is simplified by adopting a single-stage network structure, which significantly improves the representation ability and estimation accuracy of the model for human joint point locations. The model also has good generalization performance and efficiency advantages. To effectively embed the pixel location information into the feature map, coordinate convolution [22] is used to enhance the perception of the keypoint feature location information by adding coordinate channels to the classical convolution. The coordinate attention mechanism [23] is introduced to enhance the attention allocation of the feature map to the nodal position capability. After the coordinate convolutional focuses on the position information, the feature information (such as local and global information) is extracted using lightweight residual blocks. A block is composed of small-sized convolution and smoothing activation functions that are integrated with residual connections. It reduces the computation and complexity of the feature extraction network while retaining spatial and channel features, enhancing keypoint localization and model accuracy. It also has good generalization and can be used for other computer vision tasks to balance network accuracy and efficiency. In summary, the main contributions of this paper are as follows:

- We propose an innovative model that uses pixel coordinates to guide the estimation for the position-sensitive task of 2D human pose estimation. The model leverages the intrinsic features of the image in the network design, enhances the model performance, and balances accuracy and efficiency.
- By using coordinate convolution and a coordinate attention mechanism, we can take advantage of pixel coordinate position information effectively, optimize the structure of the single-stage network, allow the network to focus on important spatial features, and enhance the spatial perception ability and estimation accuracy of the model.
- We propose a lightweight residual block that extracts key features with less computation and complexity, generalizes well, and suits various computer vision tasks.

The remaining sections of this article are structured as follows. Section 2 discusses the related work. Section 3 describes our method of pose estimation in depth. Section 4 contains the results of the experimental data and analyses. Section 5 summarizes the discussion of the estimation method of this paper.

## 2. Related Work

Advances in 2*D* multi-person pose estimation methods for single high-resolution human pose images and methods for representing critical feature information in images are as follows.

*2.1. Human Pose Estimation*

To improve the effectiveness of keypoint estimation, there are many research works in the field of pose estimation to optimize and innovate various aspects of data processing, keypoint coordinate localization, training methods, and network models.

Data augmentation can effectively improve the quality and quantity of original data, thereby enhancing the generalization ability and robustness of the model. Peng et al. [24] proposed an innovative method that leverages adversarial learning to jointly optimize data augmentation and network training, achieving significant improvements in model performance without additional data.

For keypoint coordinate localization, there are mainly two methods: direct coordinate regression [14] and Gaussian heatmap regression [25]. The former has faster training and inference speeds, while the latter enhances the spatial generalization ability of the model. Liu et al. [26] proposed a novel coordinate decoding method based on heatmap regression, which reduces the error generated during heatmap decoding and improves the accuracy of keypoint regression. Li et al. [27] proposed an innovative keypoint coordinate classification method called Simple Coordinate Classification (SimCC), which transforms the problem into two classification tasks, one for horizontal coordinates and one for vertical coordinates. SimCC has strong generalization ability and can be easily applied to various pose estimation models.

To reduce the pressure of model training and improve the convergence speed of the model, the model training method can be optimized. Zhang et al. [28] proposed a fast pose distillation model training method, which transfers latent knowledge from a previously trained larger teacher model to a smaller target pose model, achieving fast and efficient pose estimation. Dai et al. [29] proposed a new regression cross-entropy (RCE) loss function, aiming to accelerate convergence and improve accuracy.

In terms of improving the network structure, the top-down human pose estimation method first uses a detector to detect the bounding box of all people in the image and crop it to a single-person image to quickly and accurately narrow the keypoint estimation. Then, it uses a base pose estimation model to locate the key points, and the post-processing part predicts the final keypoint coordinates of each person. Researchers usually improve the basic models for human pose estimation, which can be divided into two types: multi-stage and single-stage. Both types of models explore the depth features hidden in images, and extract and fuse semantic, global, and local features to better predict the locations of human keypoints. CPM [15] proposes an end-to-end estimation model that generates the exact joint locations step by step through a multilevel serialized network, using intermediate supervision to avoid the gradient disappearance problem. Stacked Hourglass [16] integrates human spatial relationships through multi-scale feature fusion and uses intermediate heat map supervision, improving network performance. However, the accuracy is constrained by the depth of the model. Then, Regional Multi-Person Pose Estimation (RMPE) [17] proposed a symmetric spatial transformer network to extract high-quality single human regions in the inaccurate bounding box and parametric pose no-maximum suppression to solve the pose redundancy detection problem. CPN [18] proposed a two-stage network based on a feature pyramid, which can detect easy and difficult keypoints, respectively, and better integrate multi-scale feature information. On this basis, MSPN [19] made structural improvements by reusing the pyramid structure to highlight the focus on features. HRNet [20] designed several parallel branches with different resolutions to maintain high-resolution features and was able to fuse features from different scales with each other, combining local and global semantic information to achieve more accurate pose estimation. The multi-stage model has a complex and repetitive structure, and the extracted features may be duplicated and redundant, with low computational efficiency and a certain degree of resource waste. To balance efficiency and accuracy, Xiao et al. [21] proposed a single-stage pose estimation network—Simple Baseline—as a simple and effective network that uses a deconvolution module after the backbone to recover the image size and predict the heatmap without using jump connections. The simple network architecture achieves good

performance. However, the above multi-stage and single-stage models were investigated to explore the image's feature information, ignoring its unique properties. High-resolution 2*D* human pose images imply high-precision pixel coordinate position information, and the relative position relationship of the image itself has yet to be fully exploited explicitly.

The deep-learning-based human pose estimation task mainly uses convolutional neural networks to learn the feature representation of human images. The unique properties of the image itself actually imply some key information, and for high-resolution human pose images, which implicitly contain high-precision coordinate location information, existing human pose estimation methods have not fully exploited this location information explicitly. Therefore, this paper further optimizes the network model in terms of combining the coordinate information implied by the images.

### 2.2. Feature Extraction

Feature maps are often represented and learned by convolutional neural networks. The initial deep convolutional network models contain convolutional, pooling, and fully connected layers to extract information on the details, structure, and semantics in images at different levels of abstraction. However, feature learning needs to be improved due to the few convolutional layers in the initial models. Later, the number of network layers and structures was continuously explored to enhance the characterization of feature maps. The Visual Geometry Group (VGG) [30] used multiple small-scale convolutional kernels instead of large-scale ones. It demonstrated that increasing the network depth using small-scale convolutional kernels could improve the performance to a certain extent, and it can reduce the error rate and offer solid generalization ability. However, it consumed more computational resources, and the fully connected layers occupied more memory and did not significantly help the performance. Therefore, the Deep Residual Network (ResNet) [31] discarded the fully connected layer and deepened the network depth further. However, it was found that the gradient problem occurred when back-propagating the overly deep network, so the residual structure was proposed to fuse the before and after convolutional features through a short-circuit link. The Dense Convolutional Network (DenseNet) [32,33] proposed a more densely connected mechanism. It achieves dense connections between layers through dense blocks. Each layer is directly connected to all other layers with the same feature map size. It can enhance feature propagation, reduce the number of parameters, and reduce the risk of gradient disappearance. However, DenseNet also brings a large amount of computation and memory consumption. Darknet [34] also fused the idea of residuals to avoid the gradient problem caused by deepening the network and used convolution with a step size of two instead of pooling and the network structure of full convolution to enhance feature expression. In convolutional neural networks, it is vital to balance accuracy and efficiency to improve feature learning expression while reducing unnecessary resource waste. Enhancing the feature learning capability of the network can be applied by full convolution, using small-scale convolutional kernels to maintain the image properties and compress features, which is convenient to reduce the waste of resources and the number of parameters. Combining the residual structure can avoid the gradient problem and facilitate the fusion of features before and after convolution to highlight the critical features.

Regardless of the type of convolutional neural network, the convolutional layer is the crucial component. In human pose estimation, a suitable convolutional computation will be used in a targeted manner for different purposes. Depthwise separable convolution can be used to reduce the number of parameters and achieve a lightweight human pose estimation network [35–37]. It requires some accuracy but significantly reduces the algorithm's complexity and is suitable for real-time estimation on mobile devices. To focus on the semantic mapping connections between human keypoints, the topological relationships implicit in the graph are learned using semantic graph convolution [38] to capture the local and global semantic relationships of the keypoints, which helps to optimize the global state for enhancement. Pyramid convolution [39] can effectively extract multi-scale pose

feature information, adapt to human body recognition tasks of different sizes in images, and expand the receptive field. It achieves efficient feature extraction and enhances the feature expression ability while keeping the parameters and computation unchanged. Atrous convolution [40] can expand the receptive field and improve the target keypoint detection effect without reducing the image resolution and losing detailed information. At the same time, it can also reduce the parameters and increase the computation efficiency. The abovementioned convolutional approaches have different advantages. However, none of them achieves the high-precision position representation of human keypoints by convolution without focusing on and optimizing the position information from the convolutional level. For the keypoint estimation problem, convolutional feature information extraction while focusing on location change is a fast and effective enhancement method.

In terms of enhancing feature representation, attention mechanisms can increase the corresponding weight coefficients according to the importance of different feature information in each layer of the network to highlight more essential information. In pose estimation, multiple types of attention are used to improve the accuracy of the model. These include multi-resolution attention, which generates attention maps at different scales to enhance features, and multi-semantic attention, which encodes attention maps with different semantics. Hierarchical attention is used to encode the overall human body and local node scaling in the high and low layers, respectively. Together, these multiple attentions enhance the context modeling and overall estimation of the model [41]. In order to fuse local and global information of the visual scene, the global context attention module is combined with pyramidal convolution [42] to exchange a slight increase in model parameters for a practical improvement in model performance. The global context attention mechanism captures the contextual relationships between keypoints, integrates the human body structure into the network [43], generates more accurate heatmaps, and detects more invisible human keypoints, thus further improving the model's performance. The spatial channel hybrid attention module is used to enhance the fusion of different scales of information in the network. At the same time, the self-attention mechanism is used to model the long-range dependencies between global regions and to improve the ability to locate "hard" keypoints by using other "easy" keypoints. It improves the performance of the overall structure [44]. The combination of a convolutional block attention module and residual modules allows the network to focus more on useful feature information [45], replacing the down-sampling pooling layer in the network with the combined module and preserving the joint positions, thus solving the problem of losing key feature information. The attention mechanism focuses on semantic, channel, global, and local information that can enhance the feature learning capability. Human pose estimation aims to identify the locations of keypoints; then, focusing on coordinate positions through the attention mechanism can also effectively improve the estimation accuracy. For human pose estimation, the flexible use of position-dependent convolution and attention benefits the high-accuracy position representation.

## 3. Method

Assume that the input feature image containing the human pose is $I$ and the dimension is $W \times H \times C$ ($W$ is for width, $H$ is for height, $C$ is for channel), and there are $K$ skeletal keypoints to be estimated. We solve the problem by first estimating $K$ heatmaps of size $W' \times H'$, $\{H_1, H_2, H_3 \cdots H_K\}$, where each heatmap indicates the position confidence of a single keypoint, and then the coordinates of the maximum point of the predicted heatmap are derived from the function. The coordinates are proportionally recovered to the original image size to obtain the final output. In this paper, we propose a human keypoint estimation method combining the pixel coordinate information of $2D$ images to achieve the high accuracy requirement of skeletal keypoint location estimation.

### 3.1. Framework

The pixel coordinate guided 2*D* human pose estimation network proposed in this paper is a single-stage network with a top-down approach. Figure 1 shows the complete network framework. The network is mainly divided into two parts: encoding and decoding. In the encoding stage, expanding the receptive field to extract high semantic information related to the human keypoints, the proposed coordinate convolution extraction method uses the embedded pixel coordinates to reduce the feature map size for information retention. For feature maps of different scales, we use different numbers of lightweight residual modules to extract deep image semantic, spatial, and channel features. To achieve a relative balance between feature extraction accuracy and efficiency, the number of the modules is $[1, 2, 8, 8, 4]$. The size of the image is reduced by half each time, the number of channels is increased by a factor of 2, and then convolution changes the number of channels to 256. In the decoding stage, using the deconvolution module is the simplest way to recover the image to the target heat map scale, and the subsequent use of $3 \times 3$ convolution can alleviate the tessellation effect of deconvolution to some extent. The image size in decoding is doubled each time up to one fourth of the original image, and the number of channels is kept constant. In the encoding and decoding stages, coordinate attention is combined to enhance the representation of the human joint location information, which can focus on the coordinate location information more effectively and thus improve the accuracy. This model uses $1 \times 1$ convolution to generate $K$ Gaussian heatmaps $H_k$ after the feature map is restored to the one-quarter scale of the original image. We generate ground truth heatmaps by centering a 2*D* Gaussian on the ground truth location of each keypoint with a standard deviation of one pixel. The sigma value is 2 and the Gaussian kernel size is $13 \times 13$, following the 3 sigma principle only for the region with three times Gaussian sigma distribution, which speeds up the calculation. A mean square error ($MSE$) loss function is used to compare the predicted heatmap with the ground truth heatmap, and the Adam optimizer dynamically adjusts the learning rate and other parameters to find the optimal solution.

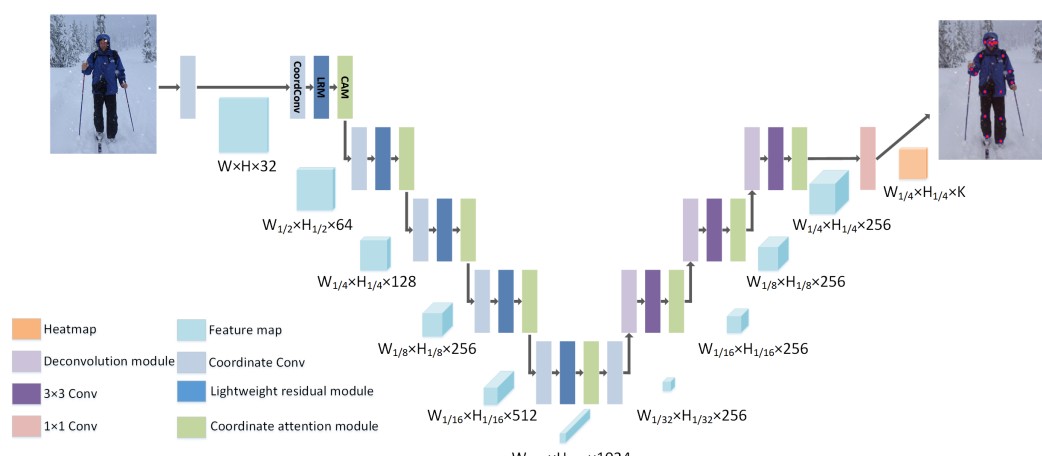

**Figure 1.** The framework of our networks. The network extracts keypoints by encoding and decoding, focusing on coordinate position information at each stage to achieve highly accurate pose estimation.

### 3.2. Coordinate Convolution

In order to improve the estimation accuracy of human keypoints in the process of pose feature extraction, the coordinate convolution with embedded pixel position information in human pose estimation is proposed, and the structure is shown in Figure 2. Firstly, the feature map $I$ with the size of $W \times H \times C$ is connected with the feature map $I'$ containing the pixel coordinate encoding information. The two channels in $I'$ represent the $X$ and $Y$ coordinates of image pixels, respectively. The feature map embedded with pixel coordinates has the spatial perception capability with the size of $W \times H \times (C + 2)$, and then the

features combining the pixel coordinate channel information are extracted by the $C^*$ group $K \times K \times (C+2)$ convolution kernel. Finally, we concatenate each group of output features $z_i$ to obtain the target feature map $Z$ with the size of $W^* \times H^* \times C^*$, and the formula is expressed as follows [22]:

$$z_i = F_i\left(\left[I_{(W \times H \times C)}, I'_{(W \times H \times 2)}\right]\right) \qquad (1 \leq i \leq c^*) \qquad (1)$$

$$Z_{(W^* \times H^* \times C^*)} = [z_1, z_2, z_3, \cdots z_{c^*}] \qquad (2)$$

where $I_{(W \times H \times C)}$ denotes the input feature map, $I'_{(W \times H \times 2)}$ denotes the pixel coordinate feature map, $[\cdot, \cdot]$ denotes the channel-level concatenation operation, $F_i$ denotes the $C^*$ group $K \times K \times (C+2)$ convolution, $z_i$ denotes intermediate output feature maps, $Z_{(W^* \times H^* \times C^*)}$ denotes the final output feature map.

Coordinate convolution not only improves the computational efficiency compared with ordinary convolution but also focuses on learning spatial coordinate information according to the demand of human pose estimation for the high-precision position. It improves spatial perception by extracting rich semantic features while combining spatial global and local position relations, which facilitates the improvement of performance metrics. In human pose estimation, more spatial location information is obtained at the cost of a tiny number of computational parameters. The coordinate convolution with embedded pixel location information can extract a large amount of semantic feature information and spatial coordinate location information, which helps to estimate the locations of keypoints of the human body.

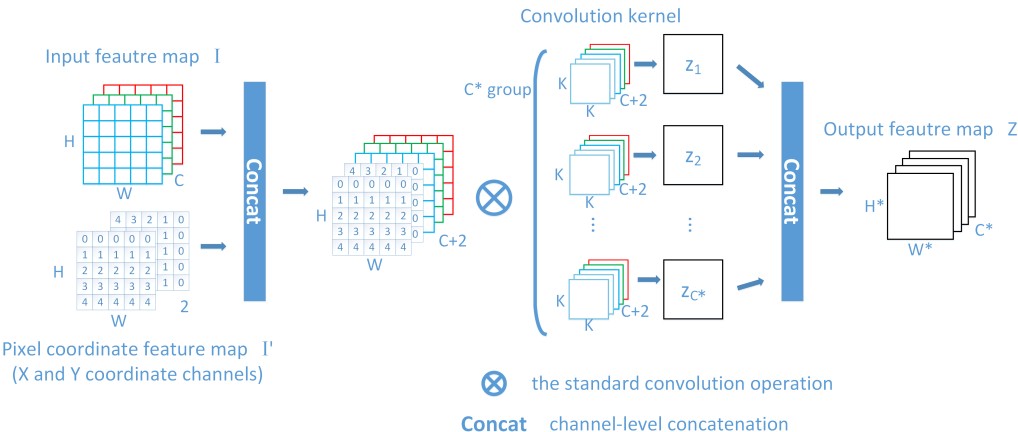

**Figure 2.** The architecture of the coordinate convolution.

### 3.3. Coordinate Attention

In order to better enhance the feature representation in human pose estimation, the attention mechanism can be applied to increase the corresponding weighting factors to highlight the location information of keypoints. The attention mechanisms can be divided into channel attention, which transforms the input into a single feature vector by 2D global pooling, and spatial attention, which transforms the input into a single channel by channel pooling. Unlike the above two attention mechanisms, coordinate attention embeds spatial location information into channel attention to focus on both channels and spatial locations. Channel attention encodes 1D features along the $X$ and $Y$ directions, respectively. This approach can capture long-range dependencies along one spatial direction to focus on the locations of human keypoints and their intrinsic connections in that direction. Meanwhile, it can retain accurate location information along the other spatial direction to prevent feature maps from being shifted in the weight generation process. Finally, it can generate a pair of direction-aware and position-sensitive attention maps and enhance the network's ability to express the coordinate position information of human keypoints through residual

connections. The coordinate attention generation weight coefficients can be divided into two steps, coordinate information embedding and coordinate attention generation [23], and the structure is shown in Figure 3.

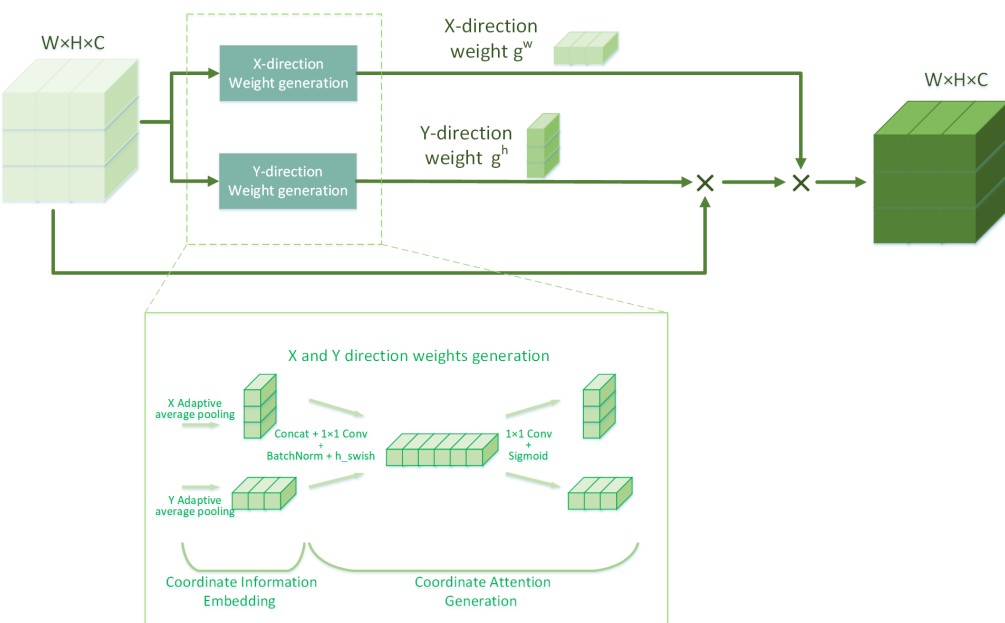

**Figure 3.** The architecture of the coordinate attention.

To better preserve the spatial location information of the feature maps, coordinate information is embedded into the channels. The features are globally adaptively averaged and pooled along the *X* and *Y* directions to encode each channel separately, which can obtain direction-aware feature maps. Furthermore, capturing the dependence of long-distance features in a single direction is beneficial for locating the target keypoint regions precisely. The coordinate attention weights are generated using the features with high spatial location information generated in the coordinate embedding stage. It combines the relationship between channels to generate channel weight coefficients $g^w$ and $g^h$ in the *X* and *Y* directions to reweight the input feature map $x_c$. The weights in both directions can focus on the location and channel, strengthen the target region features, enhance the network's ability to estimate human keypoints, and output the final feature map $y_c$.

The number of channels is compressed by concatenating the aggregated feature map and performing a $1 \times 1$ convolution on the feature map, which can be expressed as [23]:

$$f = \delta(F([z^h, z^w])) \tag{3}$$

where $[\cdot, \cdot]$ denotes the two directional feature maps for the concatenation operation; $z^h$ and $z^w$ are the 1D feature vectors output after global average pooling in the two directions, respectively; *F* represents the convolution; $\delta$ represents the batch normalization and hard_swish activation function; and *f* represents the intermediate feature map output from this operation.

Then, *f* is divided into two independent tensors $f^h$ and $f^w$ along different spatial dimensions, which are recovered into a tensor with the same number of channels as the input *X* by $1 \times 1$ convolution, and the attention weights are mapped to the range $(0, 1)$ by the sigmoid function, and the output can be expressed as [23]

$$g^h = \sigma(F_h(f^h)) \tag{4}$$

$$g^w = \sigma(F_w(f^w)) \tag{5}$$

where $F_h$ and $F_w$ are $1 \times 1$ convolution operations, $\sigma$ denotes the sigmoid function, and $g^h$ and $g^w$ are attention weights. The output features can be expressed as follows [23]:

$$y_c(i,j) = x_c(i,j) \times g_c^h(i) \times g_c^w(j) \tag{6}$$

where $c$ denotes the channel, $x_c(i,j)$ denotes the input feature map, and $y_c(i,j)$ denotes the output feature map.

*3.4. Lightweight Residual Module*

In human pose estimation, keypoints need to be identified and localized. The coordinate convolution and coordinate attention focus more on the change in position information during the convolution process, and the extraction of semantic information needs to be enhanced. Therefore, a new lightweight residual module is constructed to deepen the network depth and extract more semantic feature information, such as deeper keypoint categories and their connections, without increasing the number of parameters.

This lightweight residual block structure is shown in Figure 4. Avoiding the gradient problem caused by the deepening of the network, we use the residual structure to fuse the module input features (as shown in point *a* in Figure 4) and the features after the convolution operation (as shown in point *c* in Figure 4). Too many channels will increase the number of parameters, and the feature extraction capability has not been significantly improved, resulting in the unnecessary waste of resources. Therefore, the residual block first uses small-scale $1 \times 1$ convolution to compress the number of channels, which reduces the number of network parameters, and then uses $3 \times 3$ convolution to extract deep semantic features and restore the channel width. Batch normalization can increase the convergence speed of the network and prevent overfitting; since the linear model has limited expressiveness, the activation function can add nonlinear factors to improve the feature expression ability of the neural network model.

The nonlinear activation function enables the neural network to learn more nonlinear relationships using the Mish [46] function in the residual block, which is smoother and has no rigid zero boundaries compared to the Leaky ReLU [34] function. The images of the two activation function curves are shown in Figure 5, where the Mish activation function has no positive boundary (i.e., positive values can reach any height) and can avoid saturation due to gradient capping. The curve is smooth overall, theoretically allowing some negative values in the region of smaller absolute negative values, allowing better gradient flow and facilitating the flow of feature information. A smooth activation function allows information to penetrate deeper into the network and improves the network's learning ability, resulting in better accuracy. The expression of the Mish function [46] is

$$y = x \times tanh(ln(1 + e^x)) \tag{7}$$

The residual block balances efficiency and accuracy and can be expressed by the equations below:

$$I_{1(W \times H \times \frac{C}{2})} = \theta(F_{1 \times 1}(I_{(W \times H \times C)})) \tag{8}$$

$$I_{2(W \times H \times C)} = \theta(F_{3 \times 3}(I_{1(W \times H \times \frac{C}{2})})) \tag{9}$$

$$Z_{out(W \times H \times C)} = I_{(W \times H \times C)} + I_{2(W \times H \times C)} \tag{10}$$

where $I_{W \times H \times C}$ denotes the input feature map of the residual module, $Z_{out(W \times H \times C)}$ denotes the output feature map of the residual module, $F_{1 \times 1}$ denotes the $1 \times 1$ convolution, $F_{3 \times 3}$ denotes the $3 \times 3$ convolution, and $\theta$ denotes the batch normalization and Mish function.

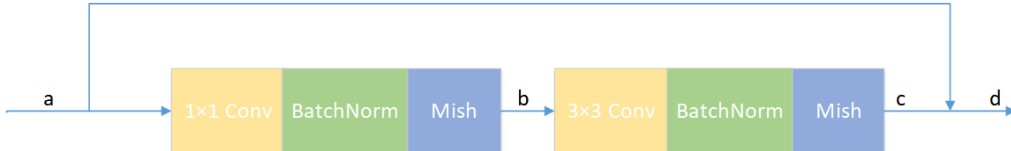

**Figure 4.** The architecture of the lightweight residual module.

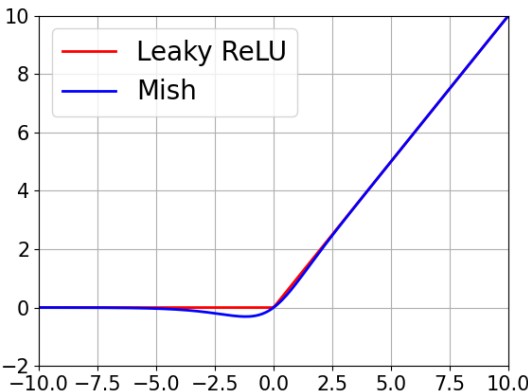

**Figure 5.** Comparison of the activation function curves between Leaky ReLU and Mish.

## 4. Experiments and Analysis

### 4.1. Dataset and Evaluation Criteria

#### 4.1.1. MPII Dataset

The experiment uses the MPII human pose dataset (MPII) [47], which is realistic and has many backgrounds, which is more conducive to simulating real situations. It has approximately $25K$ images containing more than $40K$ annotated messages with 16 $2D$ positions of human target keypoints, complete $3D$ torso and head orientations, occlusion labels, and activity labels of keypoints. Approximately $28K$ are used for training and $11K$ for testing. The data are predominantly multi-person and are mainly used as a single-frame multi-person pose test set, widely used for pose estimation and other pose-related tasks.

The percentage of correct keypoints (PCK) is a $2D$ human keypoint estimation evaluation criterion used by MPII. PCK is defined as the proportion of keypoints that are accurately detected, and the proportion of detected keypoints whose normalized distance from their corresponding ground truth values is calculated to be less than a set threshold value. The MPII normalized distance is the Euclidean distance between the predicted value of a keypoint and the ground truth, and the normalization of the human scale factor is performed using the current human head diameter as the scale factor, i.e., the Euclidean distance between the upper left point and the lower right point of the rectangular box of the head. This pose estimation metric is called PCKh. PCKh@0.5 is considered correct when the normalized Euclidean distance between the predicted value and the ground truth is less than 50% of the head size factor, and PCK is the proportion of the number of keypoints correctly predicted by this method to the total number. The PCK calculation formula is as follows:

$$PCK_{mean}^{k} = \frac{\sum_p \sum_i \delta(\frac{d_{pi}}{d_p^{def}} \leq T_k)}{\sum_p \sum_i 1} \tag{11}$$

where $i$ denotes the keypoint with ID $i$, $p$ denotes the $p$th pedestrian, $d_{pi}$ denotes the Euclidean distance between the predicted value of the keypoint with ID $i$ in the $p$th person and the ground truth, $d_p^{def}$ denotes the scale factor of the $p$th person, $T_k$ denotes the manually set threshold, $k$ denotes the $k$th threshold, $T_k \in [0 : 0.01 : 0.1]$, $PCK_{mean}^{k}$ denotes the average of the PCK value under $T_k$ thresholds, and function $\delta$ indicates the number of critical points that meet the criteria for calculation.

### 4.1.2. COCO Dataset

This experiment uses the Common Objects in Context (COCO) 2017 dataset [48], which contains 80 target categories, approximately 220$K$ annotated images, and five types of annotated information, such as object detection and human keypoint detection. Among them, around 250$K$ pieces of human body information are annotated in the images, and each human body has 17 keypoints. COCO2017 is divided into the train2017 set, val2017 set, and test-dev2017 set. We train our model on the COCO train2017 dataset, including 57$K$ images and 150$K$ person instances. We evaluate our approach on the val2017 set and test-dev2017 set, containing 5000 and 20$K$ images, respectively.

Object Keypoint Similarity (OKS) is used to measure the similarity between the predicted keypoints and the ground truth.

$$OKS = \frac{\sum_i exp[\frac{-d_i^2}{2s^2k_i^2}\delta(v_i > 0)]}{\sum_i \delta(v_i > 0)} \tag{12}$$

where $i$ denotes the number of keypoints; $d_i$ denotes the Euclidean distance between the predicted keypoint $i$ and the ground truth. $s$ denotes the scale factor whose value is the square root of the area of the human detection frame, $k_i$ denotes the normalization factor for the $i$th keypoint, $v_i \in \{0, 1, 2\}$ denotes the visibility of the $i$th keypoint, 0 denotes the unlabeled keypoints, 1 denotes no occlusion and has been labeled, 2 denotes occlusion but has been labeled, and the function $\delta$ determines whether the condition holds, and here means that only the labeled keypoints are calculated.

The evaluation metric for the main comparison in this experiment is the average accuracy (AP), which is averaged across multiple IoU values [0.50 : 0.05 : 0.95]; $AP^{50}$ and $AP^{75}$ are metrics for individual IoU thresholds; $AP^M$ and $AP^L$ are metrics set based on the size of the target object area in the dataset; $AP^M$ means $32^2 < area < 96^2$, and $AP^L$ means $area > 96^2$. The average recall (AR) is an additional metric averaged over all IoUs and categories. Intersection over Union (IoU) is defined as the ratio of the intersection area of the detector prediction bounding box to the concatenated area.

### 4.2. Experimental Details

Experimental setup: CPU Intel © Core i9-9900X 3.5 GHz, GPU Nvidia RTX2080Ti (11 GB) × 4, Ubuntu 16.04 operating system, Pytorch deep learning framework.

The height and width of the human detection frame were set to a fixed aspect ratio, 4:3, and the frame was cropped from the image and resized to a fixed size of 256 × 256 for the MPII dataset and 256 × 192 and 384 × 288 for the COCO dataset. For enhancement, the MPII dataset included a random rotation of ±30° and a random scale of ±0.25, and the COCO dataset included a random rotation of ±45° and a random scale of ±0.25. Flip tests were used for both, and the half-body data were also enhanced. Using the Adam optimizer, the initial learning rate was 0.001, and a total of 200 rounds were trained, decreasing the learning rate at 90, 120, and 150 rounds, respectively, with a scale of 0.1.

### 4.3. Analysis of Results

In order to verify the necessity and effectiveness of incorporating coordinate position information in the network proposed in this paper, this section provides a detailed analysis of the performance of the network, conducts ablation experiments to analyze the performance of each module, compares the network with the classical network for human pose estimation, and visualizes the estimation results.

#### 4.3.1. Ablation Experiments

In this section, a series of comparison experiments are conducted to verify the effectiveness of the proposed network for improving the accuracy of human pose estimation. The proposed lightweight residual module, coordinate attention, and coordinate convolution are applied sequentially to the basic framework with a structure similar to Simple

Baseline50 [21] to compare the metrics of the algorithms under different network structures. The comparison results are shown in Tables 1 and 2.

**Table 1.** Ablation experiment of the  models on the MPII dataset (PCKh@0.5).

| Method | Head | Shoulder | Elbow | Wrist | Hip | Knee | Ankle | Mean |
|---|---|---|---|---|---|---|---|---|
| Simple Baseline50 [21] | 96.351 | 95.329 | 88.989 | 83.176 | 88.420 | 83.860 | 79.594 | 88.532 |
| Baseline +Lightweight residual module | 96.696 | 95.372 | 89.501 | 83.964 | 88.489 | 84.565 | 80.444 | 88.983 |
| Baseline +Lightweight residual module +Coordinate attention | 96.794 | 95.686 | 89.756 | 84.223 | 88.523 | 85.371 | 80.963 | 89.264 |
| Baseline +Lightweight residual module +Coordinate attention +Coordinate convolution | 96.862 | 95.941 | 90.029 | 84.512 | 89.008 | 86.176 | 81.837 | 89.683 |

For the MPII dataset, we mainly use the mean value of PCK as the criterion for judgment. As seen in Table 1, the lightweight residual module alone is more advantageous than Simple Baseline50 in extracting the human pose feature information. The mean value is increased by 0.451, which proves that the lightweight residual module helps the feature information extraction and improves the index. The network structure with the addition of the coordinate attention shows a 0.281 increase in mean value, showing that the network structure incorporating the coordinate attention plays a more significant role in human pose estimation. The mean value increases again by 0.419 after using the coordinate convolution, indicating that the coordinate convolution focusing on coordinate information is more suitable for the field of human pose estimation than the standard convolution and plays an important role in estimating the position of the joint points. The metrics of each joint point are consistent with the mean change, and the proposed module in this paper improves on each joint point and the mean value. The Simple Baseline has the best metrics in head recognition and the worst in ankle recognition, and our network shows a consistent pattern.

For the COCO dataset, we mainly use the AP as the criterion for judgment. As seen in Table 2, using only the lightweight residual module, compared with Simple Baseline50, 101, and 152 [21], the AP is increased by 3.4, 2.4, and 1.8, respectively. The change in the number of parameters is an increase of 13.8M and a decrease of 5.2M, 20.8M, with a FLOPs increase of 3.8G, 0.3G, and decrease of 3.0G. The light weight of this module and its efficient feature extraction capability are verified. The model with the coordinate attention only increases the number of parameters by 0.2M and the AP increases by 0.2; compared with Simple Baseline50, the AP increases by 3.6. Using the coordinate convolution increases the number of parameters by 8.7M and the FLOPs by 0.1G, and the AP increases by 0.2. Compared with Simple Baseline50, the AP value increases by 3.8, and AP is increased by 2.2 using 82% of the number of parameters of Simple Baseline152.

**Table 2.** Ablation experiment of the models on the COCO validation set.

| Method | Params | FLOPs | AP | $AP^{50}$ | $AP^{75}$ | $AP^M$ | $AP^L$ | AR |
|---|---|---|---|---|---|---|---|---|
| Simple Baseline50 [21] | 34.0M | 8.9 | 70.4 | 88.6 | 78.3 | 67.1 | 77.2 | 76.3 |
| Simple Baseline101 [21] | 53.0M | 12.4 | 71.4 | 89.3 | 79.3 | 68.1 | 78.1 | 77.1 |
| Simple Baseline152 [21] | 68.6M | 15.7 | 72.0 | 89.3 | 79.8 | 68.7 | 78.9 | 77.8 |
| Baseline +Lightweight residual module | 47.8M | 12.7 | 73.8 | 92.3 | 81.0 | 70.7 | 78.4 | 76.9 |
| Baseline +Lightweight residual module +Coordinate attention | 48.0M | 12.7 | 74.0 | 92.5 | 81.6 | 70.9 | 78.6 | 76.9 |
| Baseline +Lightweight residual module +Coordinate attention +Coordinate convolution | 56.7M | 12.8 | 74.2 | 92.5 | 82.6 | 71.4 | 78.4 | 77.0 |

The performance of the coordinate convolution in $AP^{50}$ and $AP^L$ is inadequate. When the threshold value is set to 0.5, the deviation between the bounding box and ground truth box is large, and paying attention to the coordinate change in the convolution does not significantly help the accuracy. When the effective bounding box area is large, the human body keypoint estimation range is also extensive, so the combination of pixel coordinates in the convolution is not enough to significantly improve the accuracy. The AR only has a slight improvement compared with Simple Baseline50, and there is still a gap compared with Simple Baseline101 and 152. Considering the network depth, our model is similar to Simple Baseline50, which is approximately one half and one third of Simple Baseline 101 and 152. It shows that the network depth has a more significant effect on the AR, and increasing the network depth can effectively improve the value.

In summary, the comprehensive experiments on two datasets demonstrate the effectiveness of the lightweight residual module, coordinate attention, and coordinate convolution in the network, which have improved the primary evaluation indexes and reduced a certain number of parameters and FLOPs based on improved performance indexes.

4.3.2. Comparison with Classical Models

In this section, the performance metrics of the network are compared with the classical methods on the COCO dataset, and the comparison results are shown in Tables 3 and 4.

The comparison results of our method with other state-of-the-art network methods in the COCO validation set are shown in Table 3. All methods use input images with the same resolution (256 × 192), and our model has good performance. Compared with Simple Baseline152 [21], our method improves the AP values by 2.2 and reduces the parameters and computation by 11.9M and 2.9G, respectively. Even without pre-training, our method has AP values 0.8 higher than HRNetW32 [20]. Compared with pre-trained HRNet [20], our method has a small gap, but it surpasses HRNetW48 [20] by 1.9 and 0.4 on the $AP^{50}$ and $AP^{75}$ indicators, respectively, which indicates that combining pixel coordinates can better identify and locate human keypoints within human detection boxes, retaining more spatial feature information. In addition, compared with FasterPose50 and 101 [29], our method increases by 2.5 and 0.7 the AP values, respectively. Compared with FasterPose152 [29], our method has consistent accuracy but lower computation.

**Table 3.** Comparison results of different models on the COCO validation set.

| Method | Params | Input Resolution | Params | FLOPs | AP | AP$^{50}$ | AP$^{75}$ | AP$^{M}$ | AP$^{L}$ | AR |
|---|---|---|---|---|---|---|---|---|---|---|
| 8-Stage Hourglass [16] | N | 256 × 192 | 25.1M | 14.3 | 66.9 | — | — | — | — | — |
| CPN50 [18] | Y | 256 × 192 | 27.0M | 6.2 | 68.6 | — | — | — | — | — |
| CPN50+OHKM [18] | Y | 256 × 192 | 27.0M | 6.2 | 69.4 | — | — | — | — | — |
| Simple Baseline50 [21] | Y | 256 × 192 | 34.0M | 8.9 | 70.4 | 88.6 | 78.3 | 67.1 | 77.2 | 76.3 |
| Simple Baseline101 [21] | Y | 256 × 192 | 53.0M | 12.4 | 71.4 | 89.3 | 79.3 | 68.1 | 78.1 | 77.1 |
| Simple Baseline152 [21] | Y | 256 × 192 | 68.6M | 15.7 | 72.0 | 89.3 | 79.8 | 68.7 | 78.9 | 77.8 |
| HRNetW32 [20] | N | 256 × 192 | 28.5M | 7.1 | 73.4 | 89.5 | 80.7 | 70.2 | 80.1 | 78.9 |
| HRNetW32 [20] | Y | 256 × 192 | 28.5M | 7.1 | 74.4 | 90.5 | 81.9 | 70.8 | 81.0 | 79.8 |
| HRNetW48 [20] | Y | 256 × 192 | 63.6M | 14.6 | 75.1 | 90.6 | 82.2 | 71.5 | 81.8 | 80.4 |
| PyConvHGResNet50 [42] | Y | 256 × 192 | 36.2M | 9.4 | 71.1 | 91.0 | 79.3 | 67.9 | 76.9 | 76.6 |
| FasterPose50 [29] | Y | 256 × 192 | 25.7M | 3.8 | 71.7 | — | — | — | — | — |
| FasterPose101 [29] | Y | 256 × 192 | 44.7M | 7.2 | 73.5 | — | — | — | — | — |
| FasterPose152 [29] | Y | 256 × 192 | 60.4M | 10.6 | 74.2 | — | — | — | — | — |
| Ours | N | 256 × 192 | 56.7M | 12.8 | 74.2 | 92.5 | 82.6 | 71.4 | 78.4 | 77.1 |

**Table 4.** Comparison results of different models on the COCO test set.

| Method | Params | Input Resolution | Params | FLOPs | AP | AP$^{50}$ | AP$^{75}$ | AP$^{M}$ | AP$^{L}$ | AR |
|---|---|---|---|---|---|---|---|---|---|---|
| OpenPose [8] | — | — | — | — | 61.8 | 84.9 | 67.5 | 57.1 | 68.2 | 66.5 |
| HigherHRNet [10] | — | — | — | — | 65.9 | 86.4 | 70.6 | 73.3 | 66.5 | 57.9 |
| CPN [18] | — | 384 × 288 | — | — | 72.6 | 86.1 | 69.7 | 78.3 | 64.1 | — |
| RMPE [17] | — | 320 × 256 | 28.1M | 26.7 | 72.3 | 89.2 | 79.1 | 68.0 | 78.6 | — |
| Simple Baseline50 [21] | Y | 256 × 192 | 34.0M | 8.9 | 70.0 | 90.9 | 77.9 | 66.8 | 75.8 | 75.6 |
| Simple Baseline152 [21] | Y | 384 × 288 | 68.6M | 35.6 | 73.7 | 91.9 | 81.1 | 70.3 | 80.0 | 79.0 |
| HRNetW32 [20] | Y | 384 × 288 | 28.5M | 16.0 | 74.9 | 92.5 | 82.8 | 71.3 | 80.9 | 80.1 |
| HRNetW48 [20] | Y | 384 × 288 | 63.6M | 32.9 | 75.5 | 92.5 | 83.3 | 71.9 | 81.5 | 80.5 |
| PyConvHGResNet50 [42] | Y | 256 × 192 | 36.2M | 9.4 | 71.1 | 91.0 | 79.3 | 67.9 | 76.9 | 76.6 |
| FasterPose50 [29] | Y | 256 × 192 | 25.7M | 3.8 | 70.8 | 91.3 | 78.8 | 67.2 | 76.8 | 76.4 |
| Ours | N | 256 × 192 | 56.7M | 12.8 | 73.2 | 91.5 | 81.5 | 70.0 | 79.0 | 78.8 |
| Ours | N | 384 × 288 | 56.7M | 28.8 | 75.1 | 92.4 | 83.0 | 71.5 | 80.4 | 79.4 |

The comparison results of our method with other state-of-the-art network methods on the COCO test set are shown in Table 4. Our method achieves an AP value of 75.1 without using pre-trained models and extra training data. It can be seen from the table that the higher the input image resolution, the better the model effect and the more accurate

keypoint localization. The experimental results on the test set are consistent with those on the validation set. Compared with the Simple Baseline [21], PyConvHGResNet [42], and FasterPose [29] models, our model has a better estimation effect. Our model always pays attention to position information and obtains similar results to HRNet [20] with lower parameter amounts without using pre-trained models, which shows that paying attention to spatial position information is very necessary for human pose estimation.

Compared to the bottom-up methods, such as DeeperCut, OpenPose, Associative Embedding, PersonLab, MultiPoseNet, PifPaf, and HigherHRNet, summarized in [1,2], the method proposed in this paper shows a significant improvement in accuracy. Bottom-up approaches estimate the keypoints of all individuals in the image and assemble them into human bodies, which can result in substantial errors. In contrast, our proposed method first narrows the estimation range by detecting single-person images using a detector and then estimates the keypoints, which can alleviate the errors caused by an extensive estimation range. Moreover, our method utilizes pixel coordinate information, which performs better in small-range scenarios. Compared to the top-down approaches, such as AlphaPose, Mask-RCNN, G-RMI, CPN, RMPE, Integral Pose Regression, CFN, SimpleBaseline, and HRNet, summarized in [1,2], the proposed method effectively utilizes the ignored pixel coordinate information. Our experiments have shown that incorporating pixel coordinate information is helpful for pose estimation tasks.

Previous work on network structure design has mainly focused on how to obtain high semantic information from low-resolution images [16–19,21,42] while neglecting the loss of spatial features during the image resolution reduction. We address the correspondence of spatial features across different image resolutions by pixel location. We propose a simple encoder–decoder network architecture that uses the pixel coordinate position information contained in the image's characteristics to guide the precise localization of human keypoints. Our method has three main advantages. First, we embed pixel positions into the feature images by coordinate convolution, which leverages the relative coordinate system provided by the pixel coordinate channels. It helps to restore the pixel positions corresponding to the features when the feature image size changes, and to avoid the problem of feature information position offset. Second, our method applies coordinate attention to enhance the directional characteristics of the feature image channels, which efficiently integrates spatial coordinate information. Finally, we build the backbone network with lightweight residual blocks, which reduces the model's computational cost and parameters while balancing accuracy and efficiency. The experimental results demonstrate that incorporating coordinate information can effectively improve the model performance, especially when the input feature size is large.

However, our network also has some limitations. When the feature image resolution remains unchanged (e.g., HRNet [20]), the pixel coordinate positions provide limited spatial constraints. Moreover, the current method is only optimized in terms of network structure design; previous work has also improved the data pre-processing, key point localization, and training methods [24,26–28], which can also enhance the recognition accuracy. In the future, we plan to extend our work in the pose domain and explore how to combine pixel coordinate information with image pre-processing and coordinate decoding to achieve higher estimation accuracy.

### 4.3.3. Visualization of Experimental Results

To demonstrate the effectiveness of our model in pose estimation more intuitively, we have visualized the estimation results of different network models and selected some representative images in the COCO and MPII datasets for further demonstration and explanation, as shown in Figure 6. To better distinguish the differences, we have highlighted them with yellow circles. Simple Baseline [21] may have a more significant deviation in identifying densely populated keypoints, but combining pixel coordinate position information can somewhat alleviate the problem of keypoint offset. When the image is blurry, Simple Baseline [21] may not be able to accurately estimate the position of the human body keypoints, resulting in

missing some keypoints and distorting the human body posture or facial structure features. Our model can better restore the corresponding positions of occluded keypoints by using pixel position information and identifying more keypoints. Simple Baseline [21] cannot predict the position of occluded keypoints when an occlusion occurs. In contrast, our model can better predict the position of occluded keypoints by using the relative position relationship of pixel coordinates. In addition, our model achieves excellent performance on two public datasets, demonstrating its generality and applicability.

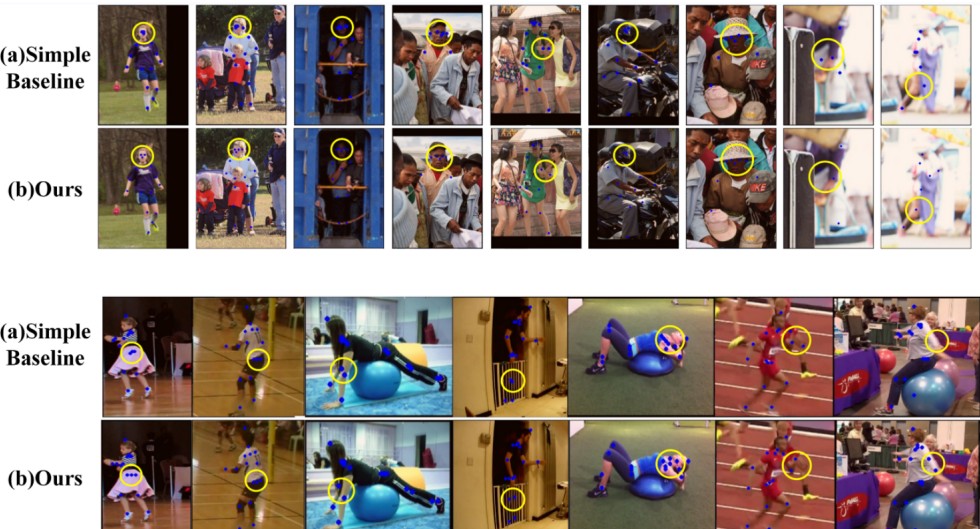

**Figure 6.** Visual comparison results of different models. The COCO dataset results are at the top, and the MPII dataset results are at the bottom. (**a**) Simple Baseline has poor recognition results under dense key points, blurred images, and occlusion. (**b**) Our approach mitigates these problems by incorporating pixel-coordinate position information.

In Figure 7, we give more comparison results and combine all keypoint connections into a complete human pose. We also show more qualitative results in real-world scenes, as shown in Figure 8. It demonstrates that our model can achieve satisfactory results in many real-world scenarios.

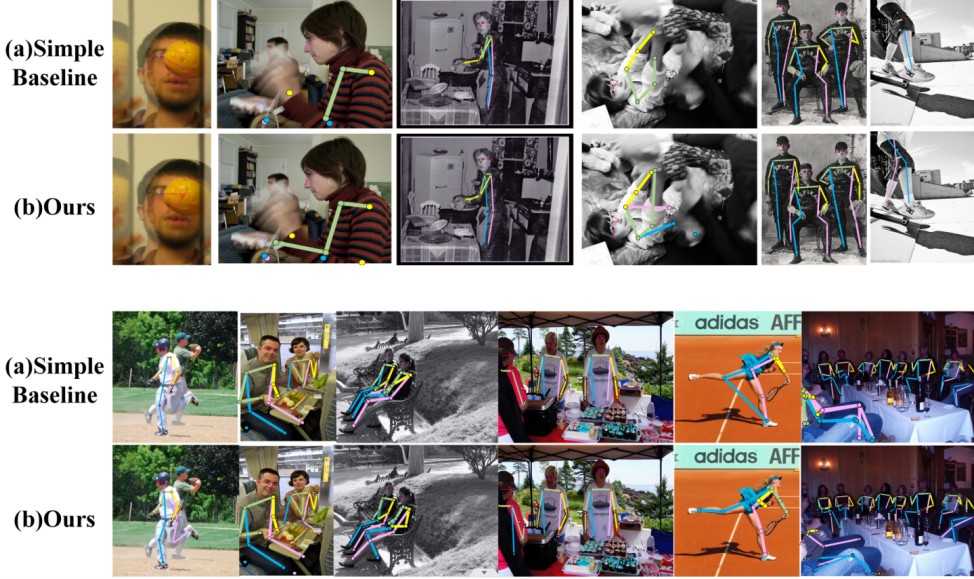

**Figure 7.** More visual comparison results. As can be seen from the figure, our method can estimate the human posture more accurately when the image is blurred or blocked. In addition, our model performs better in both single-person and multi-person scenarios.

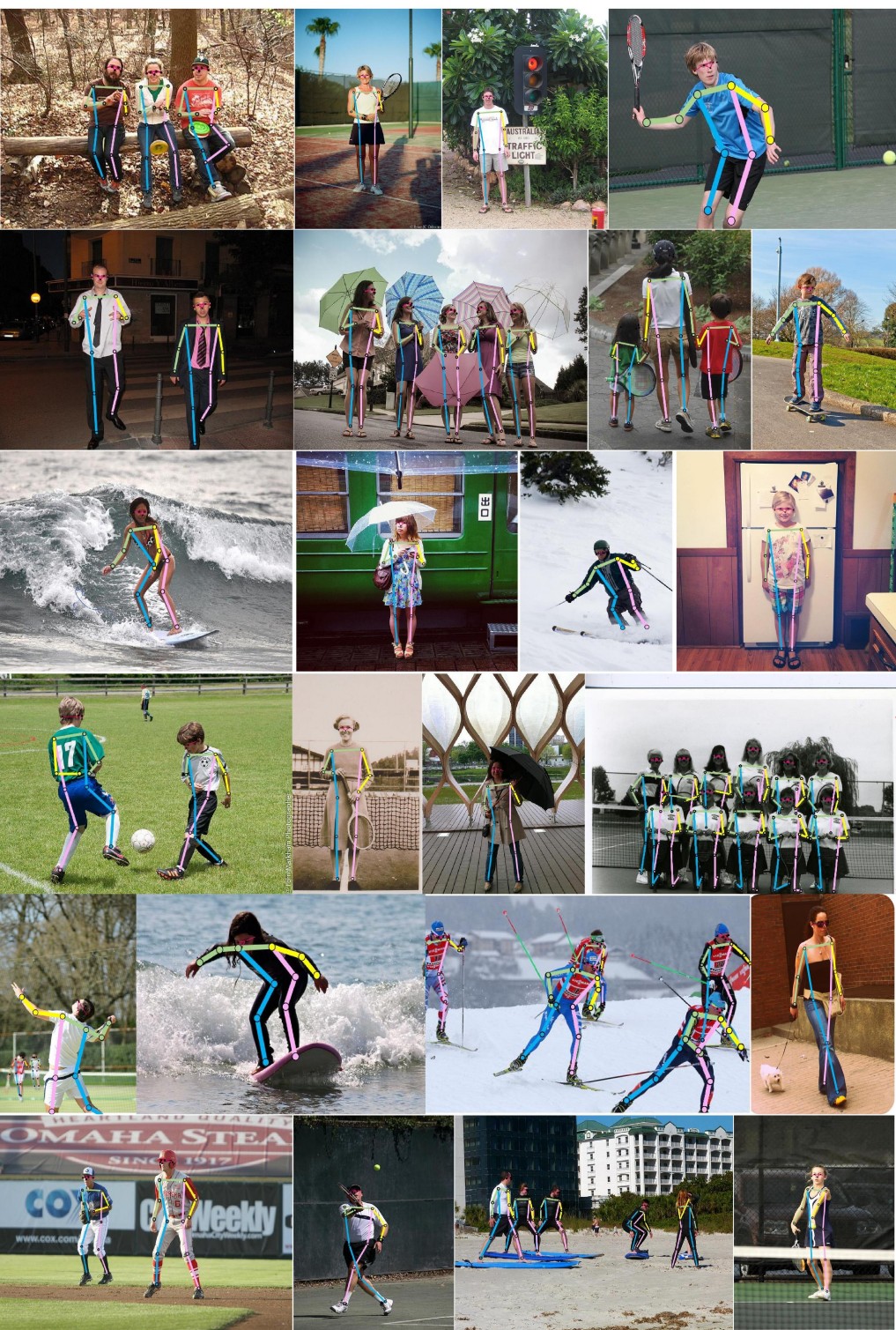

**Figure 8.** Qualitative estimated results of our model in more realistic scenarios.

## 5. Conclusions

In this paper, we propose a network that coordinates position information using a top-down framework for 2*D* human pose estimation. The model is sensitive to the coordinate position when learning the feature map information, which improves the accuracy of estimating the locations of keypoints on the human body. The network proposed in this paper effectively utilizes the high-precision pixel coordinate position information

embedded in high-resolution images. It embeds the position information in the network from several aspects, embedding coordinates in the feature maps during the feature extraction stage so that the images pay more attention to coordinate position correspondence when they change at different scales. The encoding and decoding stages, combined with the coordinate attention mechanism, always focus on coordinate information. Using the lightweight residual module as the fundamental part of the network can reduce the number of parameters based on improving the performance. The experimental results show that the proposed model outperforms the existing single-stage models in both the MPII and COCO datasets and reduces the number of parameters and the FLOPs. The results of experiments show that the approach combining coordinate location information improves the performance metrics. Our future work aims to achieve more efficient and accurate human pose estimation by optimizing the following aspects: enhancing the expression of spatial location information with data augmentation and coordinate decoding; simplifying the model complexity with network structure optimization; reducing the model training difficulty with knowledge distillation and other techniques.

**Author Contributions:** Methodology and writing—original draft preparation, X.S. and X.G.; writing—review and editing, X.G. and Q.L.; resources, X.S.; data curation, R.Z. and X.S.; supervision, X.G. and M.J.A.; project administration, X.G.; funding acquisition, Q.L. All authors have read and agreed to the published version of this manuscript.

**Funding:** This research was supported by the National Natural Science Foundation of China under Grant No. 61471263, 61872267 and U21B2024; the Natural Science Foundation of Tianjin, China, under Grant 16JCZDJC31100, and the Tianjin University Innovation Foundation under Grant 2021XZC-0024.

**Institutional Review Board Statement:** Not applicable.

**Informed Consent Statement:** Not applicable.

**Data Availability Statement:** https://cocodataset.org/#download and http://human-pose.mpi-inf.mpg.de/ (accessed on 1 March 2023).

**Conflicts of Interest:** The authors declare no conflict of interest.

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
