# Peer review of "Pixel-Coordinate-Induced Human Pose High-Precision Estimation Method"

_electronics, doi:10.3390/electronics12071648_

Round 1

Reviewer 1 Report

1. The submitted article has two corresponding authors? Is it suitable?

2. The references cannot support the contribution of this submitted article. The newer, more valuable and credible references are roughly [1], [2] and [38]. The comparison with these three literatures should be emphasized in the presentation of simulation results. The other comparisons can be simplified and not take up too much space, so as to be easy for the reader to read, and at the same time not be so cluttered.

3. Please indicate the source of the index for the derivation of the mathematical formula.

4. For the calculation algorithms of W, H, and C, please clearly present the corresponding mathematical formulas.

5. There are too many tables and too few pictures. Please simplify the tables, add 2D recognition result pictures, and explain the differences and correctness.

6. Please increase the references of SCIE journals in the past 5 years, and shorten the references of conference papers.

7. In the description in section 4.4.3, use (a)(b)(c) followed by (1)(2)...(5), it looks messy and not neat, please unify the marking method and format.

8. There are too many abbreviations. For important abbreviations, please list a summary list of abbreviations for easy reference.

Reviewer 2 Report

1. Reference: All References must be cited.

2.Some of the references for instance ref [8], and ref [9], and ref [25] are outdated try to incorporate recent articles.

3. The result and conclusion sections do not discuss the findings of the current work concerning the previous work.

4. The sole contribution of the current work in relation to existing work is not clear, authors are required to clearly indicate the contribution or scientific merits of their work.

Reviewer 3 Report

The article presents a 2D human pose estimation network with high-precision control of pixel coordinates. The article presents 2D human pose estimation network having a pixel coordinate guided high-precision. Position coordinates are embedded into single-stage network structure, convolution, and attention module. Perhaps, this approach deserves attention. However, the authors missed many good publications mentioned below. Moreover, the experimental data obtained in section 4.3.1 are worse than existing methods.

1. Li Y., Yang S., Liu P., Zhang S., Wang Y., Wang Z., Yang W., Xia S.-T. SimCC: A simple coordinate classification perspective for human pose estimation. Computer Vision – ECCV 2022: 17th European Conference, Tel Aviv, Israel, October 23–27, 2022, Proceedings, Part VI, Oct 2022, pp. 89–106. https://doi.org/10.1007/978-3-031-20068-7_6

2. Peng X., Tang Z., Yang F., Feris R.S., Metaxas D. Jointly optimize data augmentation and network training: Adversarial data augmentation in human pose estimation. 2018 IEEE/CVF Conference on Computer Vision and Pattern Recognition, Salt Lake City, UT, USA, 18-23 June 2018, pp. 2226–2234 . doi: 10.1109/CVPR.2018.00237

3. Liu S.,·He N.,·Wang C.,·Yu H.,· Han W. Lightweight human pose estimation algorithm based on polarized self-attention. Research Square, 2022, pp. 1–19. https://doi.org/10.21203/rs.3.rs-1599154/v1

4. Zhang F., Zhu X., Ye M. Fast human pose estimation. Proceedings of the IEEE/CVF conference on computer vision and pattern recognition, 2019, pp. 3517–3526.

Thus, we do not see fundamental advantages of the proposed approach. The papers mentioned above give more attractive and interesting solutions.

I can recommend the authors to carefully read the mentioned publications, maybe more, and find a way to outperform existing methods. Perhaps using a single-stage network structure, convolution and attention module for 2D human posture estimation was enough in 2017-2018, but not in 2023. The datasets are the same, but the experimental results presented in the peer-reviewed article are worse. The proposed method must be fundamentally enforced, and the article must be rewritten.

Round 2

Reviewer 1 Report

My opinions have hardly been rigorously and substantially revised (especially opinions 2, 3, 4, 5). In the last manuscript, the three main papers I proposed were not compared and simulated in detail, and most of them were still the same as the original ones. The manuscript is still the same as the previous manuscript, and four tables are generated by a large number of comparisons with conference papers. The revised manuscript is only a text change. The table presents numbers, and there are few simulation results. How to prove that the table data is real and not fake?

Reviewer 3 Report

The authors elaborated their article in accordance with the recommendations of the reviewer. I can agree that it is impossible to quickly re-design the proposed CNN architecture, and the authors did their best to improve the first version. It will be good if the authors are inspired by outstanding recent research and offer interesting solutions in the future.

Author Response

Thank you for taking the time to review our paper. The feedback and suggestions have been very helpful to our research. With the help of comments, we have delved deeper into our study and gained a better understanding and mastery of the relevant methods. We especially appreciate the suggestions you have made, as they will be of great help to our future research work. The advice has enabled us to refine our study further and improve the quality and credibility of the paper. Thank you again for your understanding and assistance!

Round 3

Reviewer 1 Report

1. Please remove the left parenthesis on line 532.

2. In response letter mentioned fig. 6  as 'To better distinguish the differences, we have highlighted them with yellow circles. ', this sentence is useful to readers, please add it in the manuscript.
